# CBEA: Competitive balances for taxonomic enrichment analysis

**Quang P. Nguyen**[1,2], **Anne G. Hoen**[1,2‡], **H. Robert Frost**[1‡*]

**1** Department of Biomedical Data Science, Geisel School of Medicine at Dartmouth College, Hanover, New Hampshire, United States of America, **2** Department of Epidemiology, Geisel School of Medicine at Dartmouth College, Hanover, New Hampshire, United States of America

‡ These authors jointly supervised this work.
* hildreth.r.frost@dartmouth.edu

**Data Availability Statement:** There are no primary data in the paper. All code for primary analyses are available at https://github.com/qpmnguyen/CBEA_analysis and we have archived our code on Zenodo (DOI: 10.5281/zenodo.5483198) at https://zenodo.org/record/5483198#.YmmZe9pBw2x. An R

## Abstract

Research in human-associated microbiomes often involves the analysis of taxonomic count tables generated via high-throughput sequencing. It is difficult to apply statistical tools as the data is high-dimensional, sparse, and compositional. An approachable way to alleviate high-dimensionality and sparsity is to aggregate variables into pre-defined sets. Set-based analysis is ubiquitous in the genomics literature and has demonstrable impact on improving interpretability and power of downstream analysis. Unfortunately, there is a lack of sophisticated set-based analysis methods specific to microbiome taxonomic data, where current practice often employs abundance summation as a technique for aggregation. This approach prevents comparison across sets of different sizes, does not preserve inter-sample distances, and amplifies protocol bias. Here, we attempt to fill this gap with a new single-sample taxon enrichment method that uses a novel log-ratio formulation based on the competitive null hypothesis commonly used in the enrichment analysis literature. Our approach, titled competitive balances for taxonomic enrichment analysis (CBEA), generates sample-specific enrichment scores as the scaled log-ratio of the subcomposition defined by taxa within a set and the subcomposition defined by its complement. We provide sample-level significance testing by estimating an empirical null distribution of our test statistic with valid p-values. Herein, we demonstrate, using both real data applications and simulations, that CBEA controls for type I error, even under high sparsity and high inter-taxa correlation scenarios. Additionally, CBEA provides informative scores that can be inputs to downstream analyses such as prediction tasks.

## Author summary

The study of human-associated microbiomes relies on genomic surveys via high-throughput sequencing. However, microbiome taxonomic data is sparse and high-dimensional which prevents the application of standard statistical techniques. One approach to address this problem is to perform analyses at the level of taxon sets. Set-based analysis has a long history in the genomics literature, with demonstrable impact on improving both power and interpretability. Unfortunately, there is limited interest in developing new set-based

package implementation of the method can be found on GitHub (https://github.com/qpmnguyen/CBEA) and on Bioconductor.

**Funding:** Funding for HRF provided by National Institutes of Health grants R21CA253408, P20GM130454 and P30CA023108 to QPN. Funding for AGH provided by National Institutes of Health grant R01LM012723. The funders had no role in study design, data collection and analysis, decision to publish, or preparation of the manuscript.

**Competing interests:** The authors have declared that no competing interests exist.

tools tailored for microbiome taxonomic data given its unique features compared to other 'omics data types. We developed a new tool to generate taxon set enrichment scores at the sample level through a novel log-ratio formulation based on the competitive null hypothesis. Our scores can be used for statistical inference at both the sample and population levels, and as inputs to other downstream analyses such as prediction models. We demonstrate the performance of our method against competing approaches across both real data analyses and simulation studies.

This is a *PLOS Computational Biology* Methods paper.

## Introduction

The microbiome is the collection of microorganisms (bacteria, protozoa, archaea, fungi, and viruses) which co-exists with its host. Previous research has shown that changes in the composition of the human gut microbiome are associated with important health outcomes such as inflammatory bowel disease [1], type II diabetes [2], and obesity [3]. To understand the central role of the microbiome in human health, researchers have relied on high-throughput sequencing methods, either by targeting a specific representative gene (i.e. amplicon sequencing) or by profiling all the genomic content of the sample (i.e. whole-genome shotgun sequencing) [4]. Raw sequencing data is then processed through a variety of bioinformatic pipelines [5, 6], yielding various data products, including taxonomic tables which can be used to study associations between members of the microbiome and an exposure or outcome of interest.

However, there are unique challenges in the analysis of these taxonomic count tables [7, 8]. The data is sparse, high-dimensional, and likely compositional [7–9]. Even though these problems are challenging, a very approachable solution is to use set-based analysis methods, also termed gene set testing in the genomics literature [10, 11]. Aggregated variables can be less sparse, and testing on a smaller number of features can reduce the multiple-testing burden. As such, gene set testing methods have been shown to increase power and reproducibility of genomic analyses. Furthermore, through the usage of functionally informative sets defined *a priori* based on historical experiments (for example, MSigDB [12], and Gene Ontology [13]), gene set analysis also allows for more biologically informative interpretations.

A diverse set of methods has already been developed in this field. Traditional methods utilize the hypergeometric distribution to test for the overrepresentation of a gene set using a candidate list of genes screened based on a marginal model [11]. Unfortunately, these approaches are sensitive to the differential expression test as well as the chosen threshold when trying to select genes for the candidate list. Aggregate score methods, which are generally preferred [14], instead assign a score for each gene set based on gene-specific statistics such as z-scores or fold change. Of these approaches, methods such as GSEA [12] perform a test for each gene set at the population level, summarizing information across all samples. Conversely, methods such as GSVA [15] and VAM [16], generate enrichment scores at the sample level and are more akin to a transformation. In addition to being able to screen for enriched sets per sample, this strategy also allows for the flexible incorporation of different downstream analyses, such as fitting prediction models, or performing dimension reduction.

In microbiome research, even when no explicit enrichment analysis is performed, researchers often aggregate taxa to higher Linnean classification levels such as genus, family, or phylum. However, there is limited research done to extend existing set-based methods to

microbiome relative abundance data. Some software suites, such as *MicrobiomeAnalyst*, do offer tools to perform enrichment testing with curated taxon sets [17]. However, the approach used in *MicrobiomeAnalyst* is a form of overrepresentation analysis at the population level and therefore similarly sensitive to the differential abundance approach used and the p-value threshold. One of the primary challenges for adapting gene set analysis to the microbiome context is the compositional nature of the data. Sequencing technologies constrain the total number of reads, and samples are expected to have the same number of reads instead of DNA content [18, 19]. However, different samples still yield arbitrarily different total read counts [9, 20], which suggests the usage of normalization methods to allow for proper comparison of feature abundances across samples. However, microbiome data sets do not follow certain assumptions that enable the cross-application of methods from similar fields (such as RNA-seq) [18, 19]. For example, DESeq2's *estimateSizeFactors* [21] assumes that the majority of genes act as housekeeping genes with constant expression levels across samples. As such, practitioners often rely on total sum normalization to transform count data into relative proportions that sum to one [22]. Some studies have provided emprical performance evaluations supporting this normalization schema [23]. Since this approach imposes a sum constraint on the data, post normalization microbiome data sets are compositional [9], which means that the abundance of any taxon can only be interpreted relative to another. Under this scenario, log-ratio based approaches from the compositional data analysis (CoDA) literature [24] are motivated to address this issue.

Unfortunately, the standard practice for aggregating variables using element-wise summations (referred to as "amalgamations" in the CoDA literature), does not adequately address the compositional issue [25]. First, inter-sample Aitchison distances computed on original parts are not preserved after amalgamation [26]. This means that cluster analyses might show different results depending on the level of amalgamation and differ from the those computed from original variables. Second, amalgamations do not allow for comparison between sets of different sizes within the same experimental condition since larger sets will have larger means and variances. Third, considering that each taxa has specific measurement biases [25], an amalgamation-based approach would make the bias of the amalgamated variable dependent on the relative abundance of its constituents. In other words, if taxon 1 has abundance $A_1$ and bias $B_1$, while taxon 2 has abundance $A_2$ and bias $B_2$, then the bias of the aggregate variable (for example, a genus) is $(A_1B_1 + A_2B_2)/(A_1 + A_2)$ (see Appendix 1. from McLaren et al. [25]). This means that bias invariant approaches (such as analyses of ratios) would no longer be invariant when applied to amalgamated variables, as bias now varies across samples. The alternative would be to multiply the proportions rather than to sum them [26].

Here, we present a taxon-set testing method for microbiome relative abundance data that addresses the aforementioned issues. Our approach generates enrichment scores at the sample level similar to GSVA [15] and VAM [16]. We leverage the concept of the $Q_1$ competitive hypothesis presented in Tian et al. [27] to formulate the enrichment of a set as the compositional balance [28] of taxa within the set and remainder taxa using multiplication as the method of aggregating proportions [26]. This well-defined null hypothesis allows us to perform significance testing with interpretable results through estimating the empirical distribution of our statistic under the null that can also account for variance inflation due to inter-taxa correlation [29].

In the following sections, we present our approach, titled competitive balances for taxonomic erichment analysis (CBEA). First, we present the step-by-step formulation of CBEA and discuss its statistical properties. Second, we detail our evaluation strategy using both real data and parametric simulations and the methods we are comparing. Third, we present results on enrichment testing using CBEA for single samples as well as at the population level. Fourth,

we show the performance of CBEA in downstream disease prediction. Finally, we discuss our results and the limitations of our method. An R package implementation of CBEA can be installed via Bioconductor. The development version can be found on GitHub (www.github.com/qpmnguyen/CBEA).

## Materials and methods

### Competitive balances for taxonomic enrichment analysis (CBEA)

The CBEA method generates sample-specific enrichment scores for microbial sets using products of proportions [30]. Details on the computational implementation of CBEA can be found in S1 File. The CBEA method takes two inputs:

- **X**: *n* by *p* matrix of positive proportions for *p* taxa and *n* samples measured through either targeted sequencing (such as of the 16S rRNA gene) or whole genome shotgun sequencing. Usually, **X** is generated from standard taxonomic profiling pipelines such as DADA2 [5] for 16S rRNA sequencing, or MetaPhlAn2 [6] for whole genome shotgun sequencing. CBEA does not accept *X* matrices with zeroes since they invalidate the log-ratio transformation. Users can generate a dense matrix *X* using their method of choice, however, by default CBEA will add a pseudocount of $10^{-5}$ if zeroes are detected in the matrix.

- **A**: *p* by *m* indicator matrix annotating the membership of each taxon *p* to *m* sets of interest. These sets can be Linnean taxonomic classifications annotated using databases such as SILVA [31], or those based on more functionally driven categories such as tropism or ecosystem roles ($A_{i,j} = 1$ indicates that microbe *i* belongs to set *j*).

The CBEA method generates one output:

- **E**: *n* by *m* matrix indicating the enrichment score of *m* pre-defined sets identified in **A** across *n* samples.

The procedure is as follows:

1. **Compute the CBEA statistic**: Let **M** be a *n* by *m* matrix of CBEA scores. Let $\mathbf{M}_{i,k}$ be the CBEA score for set *k* and sample *i*:

$$\mathbf{M}_{i,k} = \sqrt{\frac{\sum_k A_{ik}(p - \sum_k A_{ik})}{p}} \ln\left(\frac{g(\mathbf{X}_{i,j}|\mathbf{A}_{j,k} = 1)}{g(\mathbf{X}_{i,j}|\mathbf{A}_{j,k} \neq 1))}\right) \tag{1}$$

where *g*(.) is the geometric mean. This represents the ratio of the geometric mean of the relative abundance of taxa assigned to set *k* and the remainder taxa.

2. **Estimate the empirical null distribution**: Enrichment scores represent the test statistic for the $Q_1$ null hypothesis $H_o$ that relative abundances in **X** of members of set *k* are not enriched compared to those not in set *k*. Since the distribution of CBEA under the null varies depending on data characteristics (Fig 1), an empirical null distribution will be estimated from data.

   - **Compute the CBEA statistic on permuted and un-permuted X**. Let $\mathbf{X}_{perm}$ be the column permuted relative abundance matrix, and $\mathbf{M}_{perm}$ be the corresponding CBEA scores generated from $\mathbf{X}_{perm}$. Similarly, we let $\mathbf{M}_{unperm}$ be CBEA scores generated from **X**.

   - **Estimate the correlation-adjusted empirical distribution for each set**. For each set, a fit a parametric distribution to both $\mathbf{M}_{perm}$ and $\mathbf{M}_{unperm}$. The location measure estimated from $\mathbf{M}_{perm}$ and the spread measure estimated from $\mathbf{M}_{unperm}$ will be combined as the

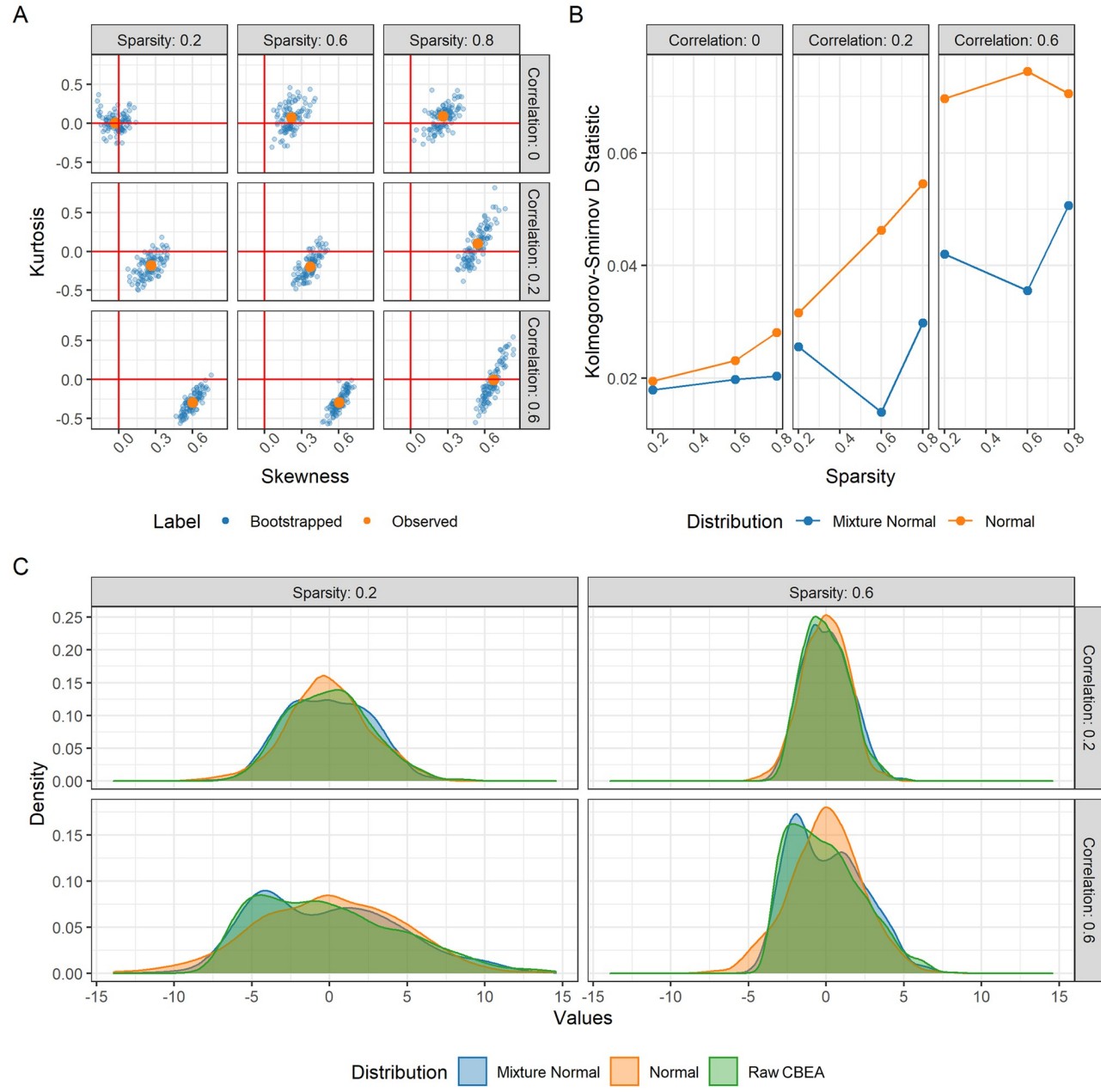

**Fig 1. Properties of the null distribution of CBEA under the global null simulations.** Panel (**B**) presents kurtosis and skewness of CBEA scores while panel (**A**) presents the goodness of fit (as Kolmogorov-Smirnov D statistic) for mixture normal and normal distributions. Panel (**C**) is a density plot of the shape of the null distribution. Results indicated the necessity of estimating an empirical null and demonstrating that the mixture distribution was the better fit compared to the basic normal.

correlation-adjusted empirical null distribution $\mathbf{P}_{emp}$ for each set. Two available options are the normal distribution and the mixture normal distribution. For the normal distribution, parameters were estimated using the method of maximum likelihood implemented in the *fitdistr* package [32]. For the mixture normal distribution, parameters were estimated using an expectation-maximization algorithm implemented in the *mixtools* package [33].

3. **Calculate finalized CBEA scores with respect to the empirical null**: Enrichment scores $E_{i,k}$ are calculated as the cumulative distribution function (CDF) values or z-scores with respect to $P_{emp}$ distribution. Raw p-values can be calculated by subtracting $E$ from 1.

## Properties of CBEA

**CBEA and balances between groups of parts.** The CBEA statistic is based on the multiplication-based aggregation approach used to calculate balances between groups of parts [26]. These balances are computed using the isometric log-ratio (ILR) transformation [30] formula. For a given balance $i$ splitting variables across sets $R$ and $S$, we have the balance coordinate $x_i^*$ as:

$$x_i^* = \sqrt{\frac{rs}{r+s}} \, \log\left(\frac{g(X_{j|j \in R})}{g(X_{j|j \in S})}\right) \tag{2}$$

where $r$ and $s$ are the cardinalities of sets $R$ and $S$ respectively, $g(z)$ is the geometric mean, and $X_j$ are values of the original predictors with indexes defined by membership in $R$ and $S$.

CBEA belongs to a set of methods that seeks to leverage compositional balances for the analysis of microbiome data [28, 34–36]. Unlike methods such as PhILR [35], CBEA does not present an orthonormal basis for the complete ILR transformation (such as a a sequential binary partition) [30]. Therefore, it is not a subclass of the ILR transformation and is adjacent to this approach. A similar method to CBEA would be phylofactor [34]. However, instead of performing an optimization procedure to identify interesting balances, CBEA constructs balances *a priori* using pre-defined sets, and formulates the enrichment of a set as the scaled log-ratio between the center of the subcomposition represented by microbes within the set and the center of the subcomposition represented by remainder taxa. This formulation aligns with the $Q_1$ null hypothesis from the gene set testing literature [27].

**Estimating the null distribution.** We can assume that the CBEA statistic, similar to other log-ratio based transforms, follows a normal distribution [30, 37]. However, when applying CBEA for hypothesis testing at the sample level, it is expected that the researcher will be testing a large number of hypotheses. Under the assumption that the number of truly significant hypotheses is low, Efron [38] showed that estimating the null distribution of the test statistic directly (termed the "empirical null distribution") is much more preferable than using the theoretical null due to unobserved confounding effects inherently part of observational studies. As such, to perform significance testing using CBEA, we also estimated the null distribution from observed raw CBEA variables.

This assumption is also supported by preliminary simulation studies (detailed below). We simulated microbiome taxonomic count data under the global null across different data features and compute raw CBEA scores as well as kurtosis and skewness of the distribution under the null in Fig 1A. We found that the characteristics of the distribution change depending on sparsity and inter-taxa correlation. Sparsity seems to drive it to be more positively skewed while inter-taxa correlation encourages platykurtic (negative kurtosis) behaviour. The effect is most dramatic under both high inter-taxa correlation and sparsity. This heterogeneity further supports the decision to estimate an empirical null distribution, as suggested by Efron [38].

Additionally, the degree of kurtosis and skewness also suggests that the normal distribution itself might not be a good approximation of the null. To address this issue, we also evaluated a two-component normal mixture distribution. Fig 1B showed the goodness of fit each distribution form using the Kolmogorov-Smirnov (KS) test statistic comparing each fitted distribution to CBEA scores generated under global null simulation scenarios. We can see that the mixture

normal distribution is a better fit (lower KS scores) than the normal distribution (across both sparsity and correlation settings).

We performed our empirical null estimation by fitting our distribution of choice and computing relevant parameters on raw CBEA scores on taxa-permuted data (equivalent to gene permutation in the gene expression literature). As such, the null distribution is characterized by scores computed on sets of equal size with randomly drawn taxa.

**Variance inflation due to inter-taxa correlation.** When taxa within a set are highly correlated, the variance of the sample mean of taxon-wise statistics is inflated. Without loss of generalizability, for a set of taxa with taxon-specific statistics $x_1, \ldots, x_p$, we have the variance of the mean $\bar{x}$ to be:

$$Var(\bar{x}) = \frac{1}{m^2}\left(\sum_{i=1}(\sigma_i^2) + \sum_{i<j}\rho_{ij}\sigma_i\sigma_j\right) \qquad (3)$$

where $\sigma_i$ is the standard deviation of taxon $i$ and $\rho_{ij}$ is the correlation between $i$ and $j$. The second term of (3) is the correlation dependent variance component, which goes to 0 if there is no correlation. The CBEA statistic follows a similar pattern. Since the geometric mean of a set of variables is equivalent to the exponential of the arithmetic mean of their logarithms, we can re-write CBEA score for a set $k$ with size $K$ as follows:

$$M_{i,k} = \sqrt{\frac{K(p-K)}{K+(p-K)}}\left(\overline{\log X_{i,j|j\in K}} - \overline{\log X_{i,j|j\notin K}}\right) \qquad (4)$$

where $p$ is the overall number of taxa, $j$ is the index of a taxa, and $K$ is the set of indices of taxa in set $k$. The CBEA statistic then looks similar to a t-statistic for difference in means of log-transformed proportions. As such, the pooled variance of CBEA is dependent on the variance inflation of both mean components $\overline{\log X_{i,j|j\in K}}$ and $\overline{\log X_{i,j|j\notin K}}$. The result of this variance inflation is an inflated type I error since highly correlated sets are also detected as significantly enriched.

However, as Wu et al. [29] showed, performing column permutations to estimate the null distribution of a competitive test statistic does not allow for adequate capture of this variance inflation factor since the permutation procedure disrupts the natural correlation structure of the original variables. It is important to address this problem since there is strong inter-taxa correlation within the microbiome [39]. Our strategy for addressing this issue is to use the location (or mean) estimate from the column permuted raw score matrix, with the spread (or variance) estimate taken from the original un-permuted scores. This still allows us to leverage the null distribution generated via column permutation while using the proper variance estimate taken from scores where the correlation structure has not been disrupted. As such, this procedure assumes that the variance of the test statistic under the alternate hypothesis is the same as that of the null. Details of the computational implementation to this estimation process can be found in S1 File.

However, set-based analysis is an exploratory approach that can help generate functionally informative hypotheses, and as such users might not want strict type I error control in favor of higher power. This is especially true for competitive hypotheses where its stricter formulation compared to the self-contained approach implies that the test naturally has lower power [11, 40]. Furthermore, sets that are highly correlated compared to background can be biologically relevant. Therefore, CBEA provides an option for users to specify whether correlation adjustment is desired.

## Evaluation

We based our evaluation strategy on gene set testing benchmarking standards set by Geistlinger et al. [41] and utilized the same approaches whenever possible. All data sets are obtained from either the *curatedMetagenomicData* [42] and *HMP16SData* [43] R packages (2020–10–02 snapshot), or downloaded from the Qiita platform [44]. All code and data sets used for evaluation of this method are publicly available and can be found on GitHub (qpmnguyen/CBEA_analysis). Additional packages used to support this analysis includes: *tidyverse* [45], *pROC* [46], *phyloseq* [47], *mia* [48], *targets* [49].

### Statistical significance

We evaluated the inference procedure of CBEA compared to alternate methods using two approaches: randomly sampled taxa sets and sample label permutation. These analyses were performed on the 16S rRNA gene sequencing of the oral microbiome from the Human Microbiome Project [1, 50]. This data set contains 369 samples split into two subsites: supragingival and subgingival. We processed this data set by removing all samples with total read counts less than 1000 and OTUs whose presence (at least 1 count) is in 10% of samples or less.

**Sample-level inference.** Due to CBEA's self-contained null hypothesis, we can perform inference at the sample level for the enrichment of a set. We evaluated this application by generating one random taxon-set of different sizes $S \in \{20, 50, 100, 150, 200\}$ across 500 iterations. Random sets can act as our estimate for type I error since they match the CBEA null hypothesis stated in Materials and methods, where we expect that within each sample sets of randomly drawn taxa should not be significantly enriched compared to the remainder background taxa. For this evaluation, we estimated type I error as the fraction of samples in which our random set is detected as significant at a p-value threshold of 0.05 with confidence bands computed from the standard error across all iterations. Additionally, this analysis also tests whether CBEA is sensitive to different set sizes.

**Population-level inference.** We can perform enrichment testing at the population level by generating corresponding sample level CBEA scores and performing a two-sample test such as Welch's t-test. In order to evaluate CBEA under this context, we generated CBEA scores of sets representing genus-level annotation in the above gingival data set [1, 50] and applied a t-test to test for enrichment (similar to GSVA [15]) across a randomly generated variable indicating case/control status (repeated 500 times). Type I error is estimated as the fraction of sets per iteration found to be significantly enriched with confidence bands computed from the standard error across all iterations. In addition, we performed a random set analysis assessment, where we generated 100 sets of different set sizes $S \in \{20, 50, 100, 150, 200\}$ and evaluated the fraction of genera that were found to be differentially abundant across the original labels (supragingival versus subgingival subsite). 95% confidence intervals were computed using the Agresti-Couli approach [51].

### Phenotype relevance

We want to evaluate whether sets found to be significantly enriched by CBEA are relevant to the research question. To perform this assessment, we relied on the gingival data set mentioned above [1, 50]. This data set was chosen because its clear biological interpretation can serve as the ground truth. Specifically, we expect aerobic microbes to be enriched in the supragingival subsite where the biofilm is exposed to the open air, while conversely anaerobic microbes should thrive in the subgingival site [52]. Genus-level annotations for microbial metabolism from Beghini et al. [53] were obtained from the GitHub repository associated with Calagaro et al. [54]. For sample-level inference, we assessed power as the fraction of

supragingival samples whereby aerobic microbes are significantly enriched. For population-level inference, power is the fraction of sets representing genus-level taxonomic assignments that were significant across subsite labels.

In addition to statistical power, we also assessed phenotype relevance through evaluating whether highly ranked sets based on CBEA scores were more likely to be enriched according to the ground truth. This is represented by the area under the receiving operator curve (AUROC/AUC) scores computed on CBEA scores against true labels (similar approach was used to evaluate VAM [16]). DeLong 95% confidence intervals for AUROC [55] were obtained for each estimate.

### Disease prediction

CBEA scores can also be used for downstream analyses such as disease prediction tasks. We utilized two data sets for this evaluation:

1. Whole genome sequencing of stool samples from inflammatory bowel disease (IBD) patients in the MetaHIT consortium [56]. This data set contains 396 samples from a cohort of European adults, of whom 195 adults were classified as having IBD (which includes patients diagnosed with either ulcerative colitis or Crohn's disease). We processed this data by removing all samples with less than 1,000 total read counts as well as any OTU that was present (with non-zero proportions) in 10% of the samples or fewer. Prior to model fitting, we back-transformed relative abundances into count data (to align the format with our 16S rRNA gene sequencing data set), using the provided total number of reads aligned to MetaPhlan marker genes (per sample).

2. 16S rRNA gene sequencing of stool samples from IBD patients in the RISK cohort [57]. This data set contains 16S rRNA gene sequencing samples from a cohort of pediatric patients (ages < 17) from the RISK cohort enrolled in the United States and Canada. Of the 671 samples obtained, 500 samples belong to patients with IBD. We processed this data set by removing all samples with less than 1,000 total read counts as well as any OTU that was present (at least 1 count) in 10% of the samples or fewer.

We evaluate disease prediction performance by fitting a random forest model [58] using CBEA scores as inputs to classify samples of patients with IBD or as healthy controls. Random forest was chosen as a baseline learner due to its flexibility as an out-of-the-box model that is easy to fit. In this instance, we evaluated predictive performance of a default random forest model (without hyperparameter tuning) AUROC after 10-fold cross validation. Additionally, we utilized SMOTE to correct for class imbalances [59]. Implementation was done using the *tidymodels* suite of packages [60].

### Comparison methods

We benchmarked the statistical properties of CBEA against existing baseline approaches. For sample-level inference analyses, we utilized the Wilcoxon rank-sum test, which non-parametrically tests the difference in mean counts between taxa from a pre-defined set and its remainder similar to CBEA. For assessments at the population level, we compared CBEA against the performance of a standard test for differential abundance with set-level features generated via element-wise summations instead. We chose DESeq2 [21] and corncob [61] because they represent methods extrapolated from RNA-seq [47] and those developed specifically for microbiome data.

Since disease prediction models and rankings-based phenotype relevance analyses seek to evaluate the informativeness of CBEA scores instead of relying on computing p-values, we

compared performance against other single-sample-based approaches from the gene set testing literature, specifically ssGSEA [62] and GSVA [15]. Additionally, for evaluating predictions, we also compared performance against a standard analysis plan where inputs are count-aggregated sets with the centered log-ratio (CLR) transformation.

## Results

In this section, we present results for evaluating statistical significance, phenotype relevance, and predictive performance. In addition to real data, we also evaluated models based on parametric simulations. Results can be found in the Supplemental Materials (S1–S5 Figs).

### Statistical significance

**Inference at the sample level.** CBEA provides significance testing at the sample level through a self-contained competitive null hypothesis. Generating random sets approximates the global null setting where within each sample, sets generated by randomly sampling taxa should not be significantly more enriched than remainder taxa.

Fig 2 demonstrates type I error of sample-level inference evaluated using the random set approach. The Wilcoxon rank sum test and unadjusted CBEA under mixture normal assumption demonstrated good type I error control at the appropriate $\alpha$ level. This fits with our expectations since the mixture normal distribution, has a much better fit than the normal distribution especially at the tails of the empirical distribution (Fig 1). However, other variants of CBEA demonstrated inflated type I error, especially correlation adjusted variants compared to their unadjusted counter parts. Encouragingly, all methods demonstrate consistent performance across all set sizes, with a slight increase in type I error at the highest levels.

Interestingly, simulation results (S1 Fig) showed an opposite pattern. Adjusted approaches were good at controlling for type I error, especially under the low inter-taxa correlation values within the set (similar to generating random sets where the natural correlation structure is disrupted). In these simulations, unadjusted approaches and the Wilcoxon rank sum test had significant type I error inflation with increasing correlation. All approaches seems to be invariant to the level of data sparsity.

**Inference at the population level.** Similar to other single-sample approaches to gene set testing such as GSVA [15], we can perform inference at the population level by utilizing a two-sample difference in means test. Here, we evaluated using CBEA scores generated under different settings with Welch's t-test in a supervised manner to assess whether a set is enriched across case/control status.

Fig 3 shows results for this scenario using both random sample label and random set evaluations. The random sample label approach (Fig 3A) provided a controlled setting where we can estimate type I error rate controlled at $\alpha = 0.05$. Across all replications, CBEA methods were able to control for type I error at the nominal threshold of 0.05, with CBEA raw scores being the most performant. Neither output types, correlation adjustment, nor distributional assumption improved performance values. Surprisingly, DESeq2 and corncob both exhibited significantly inflated type I error.

We also assessed the impact of set-size on the inference procedure by testing for enrichment using the original sample labels, but with randomly sampled sets of different sizes (Fig 3B). Overall, we observed very similar values across CBEA as well as corncob and DESeq2, suggesting that no individual method was systematically identifying too many significant sets. Additionally, similar to analogous analyses at the sample level, no approach was significantly sensitive to changes in set sizes.

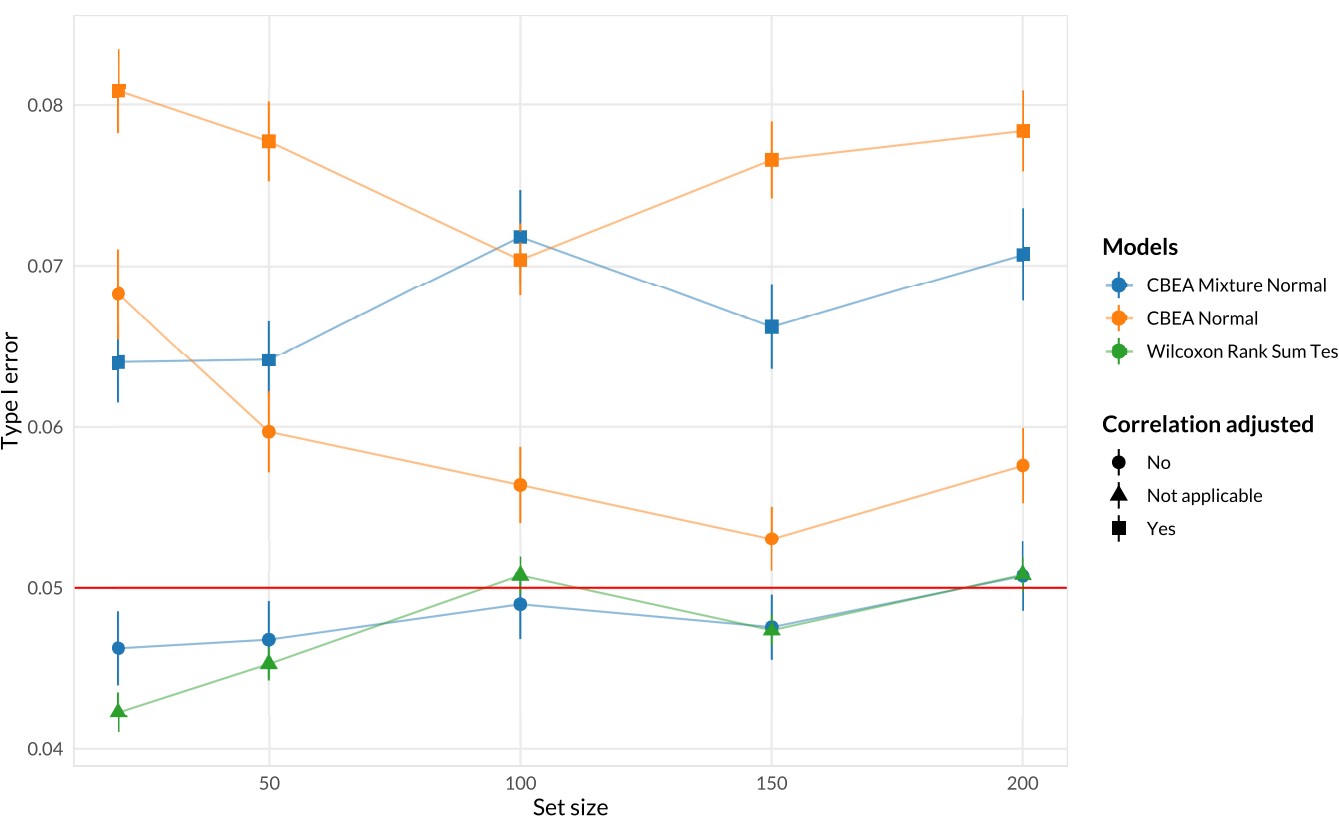

**Fig 2. Random taxa set analyses for inference at the sample level of CBEA under different parametric assumptions compared against a Wilcoxon rank-sum test.** Type I error (*y*-axis) was evaluated by generating random sets of different sizes (*x*-axis) (500 replications per size) and computing the fraction of samples in which the set was found to be significantly enriched at $\alpha$ = 0.05. Error bars represent the mean type I error ± sample standard error computed across 500 replications of the experiment. Only the unadjusted CBEA with the mixture normal distribution and the Wilcoxon rank sum test were able to control for type I error at 0.05. All approaches are invariant to set sizes.

## Phenotype relevance

**Inference at the sample level.**   In Fig 4, we evaluated whether sets found to be significant by CBEA are relevant to the phenotype of interest. We leveraged the gingival data set as stated in Properties of CBEA section knowing beforehand that aerobic microbes were more likely to be enriched in supragingival subsite samples and vice versa.

We estimated statistical power using this data set as the fraction of supragingival samples where the set representing aerobic microbes was significantly enriched. We observed that adjusted CBEA approaches demonstrated much lower power compared to the Wilcoxon rank-sum test and unadjusted variants. This is surprising given the fact that in statistical significance analyses, the adjusted CBEA approach provides inflated type I error, especially if the normal distribution assumption was chosen. This indicated a mismatch in estimating the null distribution since a high type I error did not result in increased power.

We also evaluated phenotype relevance by assessing whether enriched sets according to ground truth are preferentially ranked higher using assigned continuous scores (instead of performing a hypothesis test). This aspect was captured through computing AUROC values comparing computed enrichment scores and true labels. Consistent with the previous type I error evaluation, adjusting for correlation did not improve performance, whereas obtained AUROC were around 0.5 and at the same level as the benchmark Wilcoxon rank sum statistic.

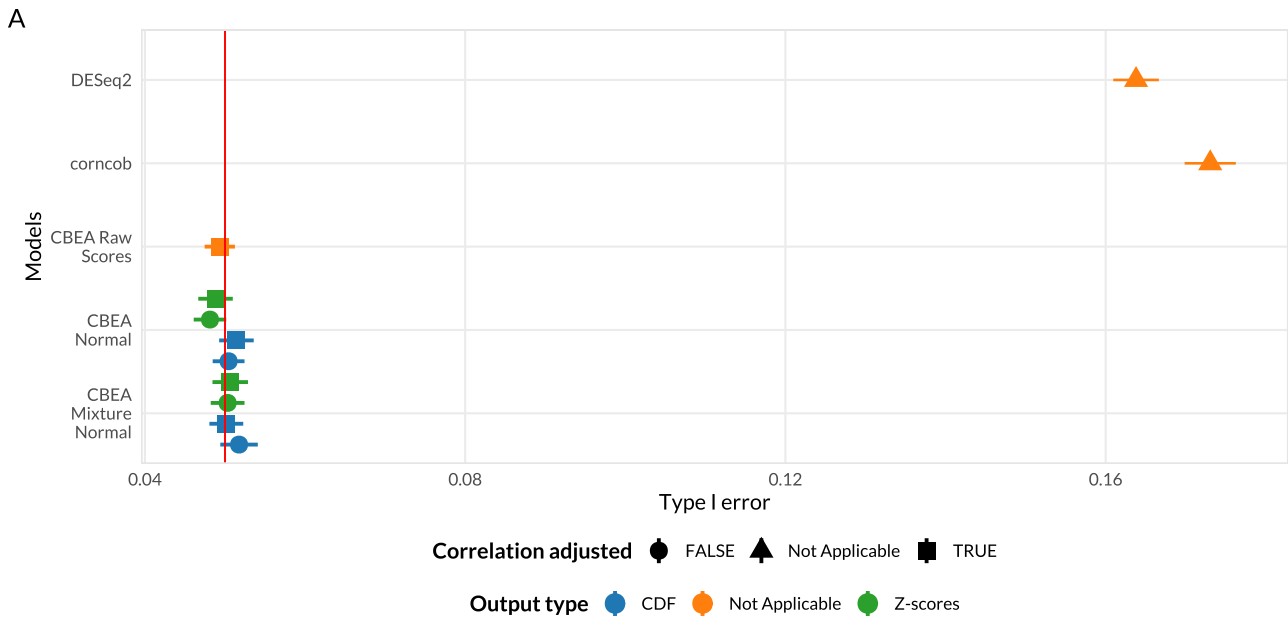

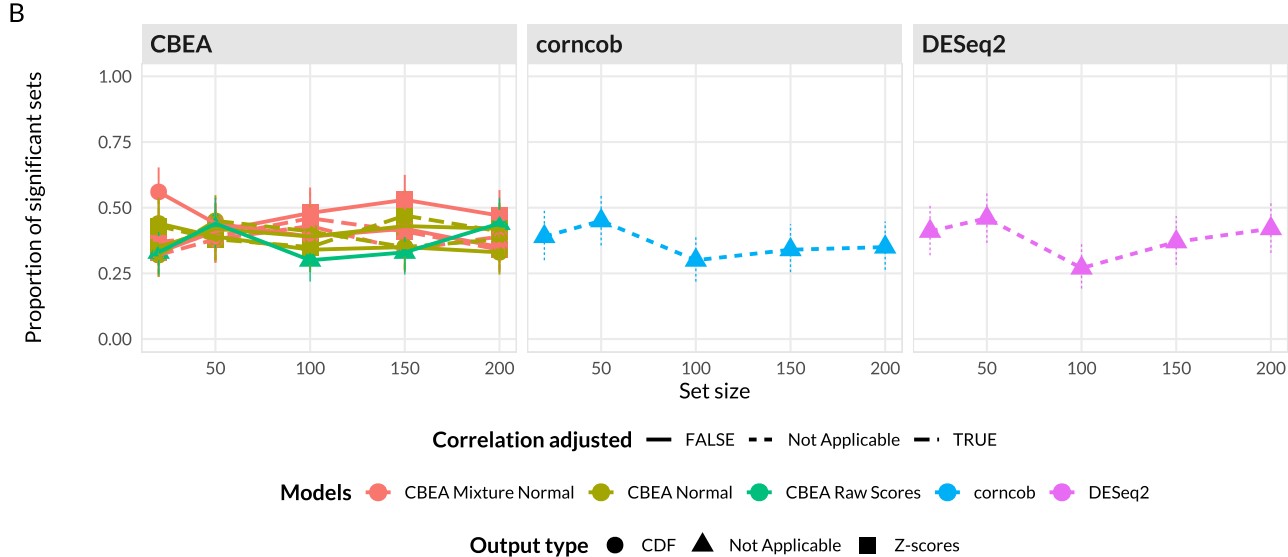

**Fig 3.** Random sample label (A) and random set (B) analyses for population level inference. (**A**) Type I error (*x*-axis) was estimated as the overall fraction of sets found to be enriched $\alpha = 0.05$ using randomly generated sample labels (500 permutations). Error bars represent the mean type I error ± sample standard error. (**B**) Proportion of significant sets (*y*-axis) using 100 randomly generated sets of different set sizes (*x*-axis). Confidence intervals computed using Agresti-Couli method for binomial proportions. For sample label permutation (**A**), all CBEA approaches were able to control for type I error but not for corncob and DESeq2. For random set analyses (**B**), all approaches demonstrated similar rates of accepting significant sets and were invariant to overall set size.

Unadjusted methods were much better at ranking true enriched sets, however the mean AUROC values were lower than alternate single-sample enrichment methods (GSVA [15] and ssGSEA [62]) even though this difference is not significant due to overlapping confidence intervals.

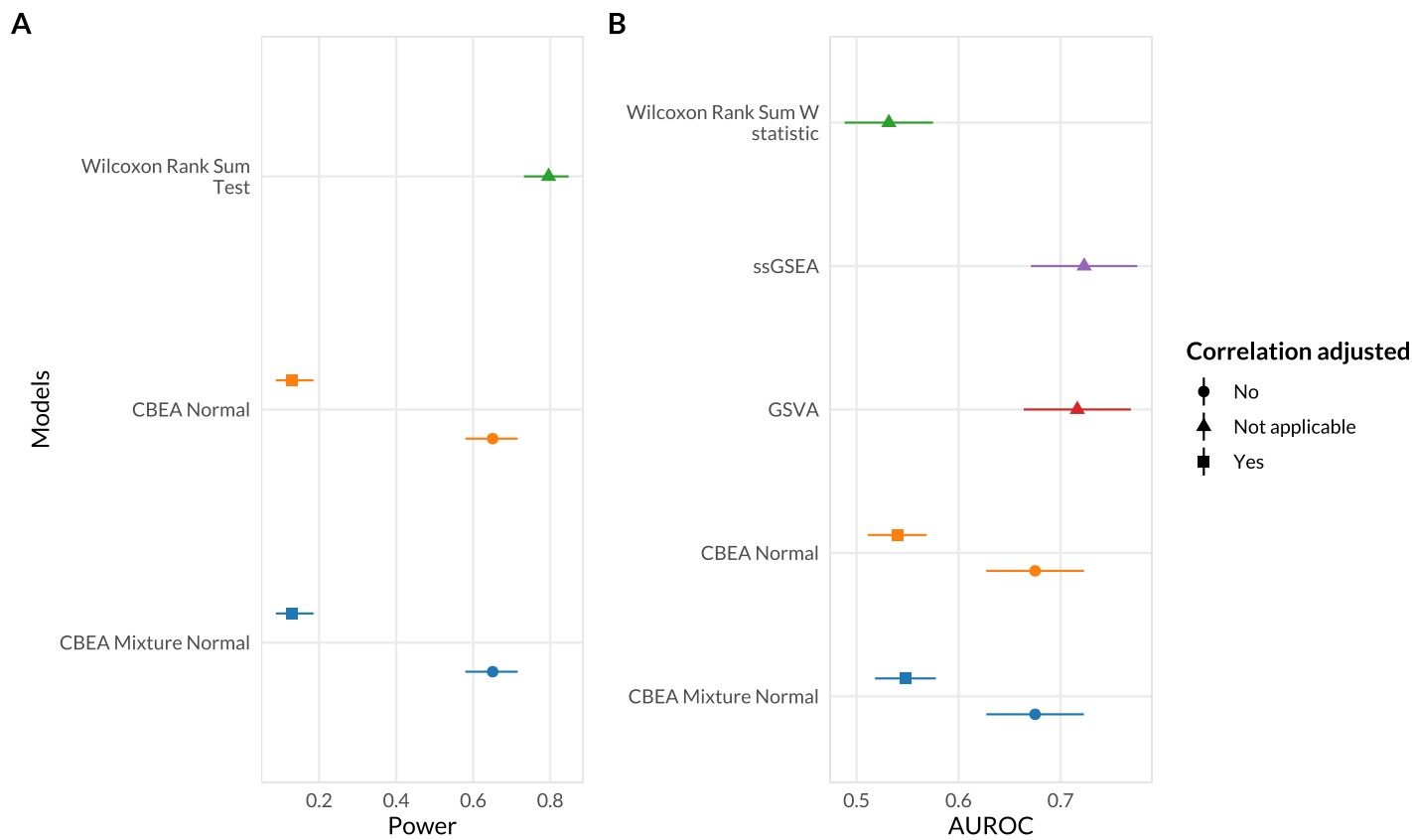

**Fig 4.** Statistical power (A) and score rankings (B) to assess phenotype relevance. (**A**) Power (*x*-axis) was estimated as the overall fraction of aerobic microbes found to be enriched in supragingival samples at $\alpha = 0.05$. 95% confidence intervals were computed using the Agresti-Couli approach for binomial proportions. (**B**) Score rankings were evaluated by comparing computed scores against true values using AUROC (*x*-axis). DeLong 95% confidence intervals for AUROC were computed.

The above results were replicated in simulation studies where we observed that adjusted approaches were very conservative and demonstrated significantly lower power (S3 Fig), with increasing correlation even at the highest evaluated effect sizes. When assessing score rankings, the performance of CBEA was closer to ssGSEA and GSVA compared to real data evaluations, however all single-sample approaches were much better than using the W statistic from the Wilcoxon Rank Sum test.

**Inference at the population level.** We also assessed statistical power for population level inference scenarios using a similar approach. Here, enrichment scores for sets representing all identified genera were computed, and power was estimated as the fraction of sets found to be differentially enriched across sample site labels (supragingival or subgingival). We compared these results against performing a differential abundance test of genus level features generated via sum-based approaches. Results are shown in Fig 5. Some CBEA variants, such as CDF outputs for the mixture normal distributional assumption, did not correctly detect as many significant sets as DESeq2 or corncob despite very close performance values. Using raw CBEA scores was best approach, however, it did not exceed values obtained from DESeq2 and corncob.

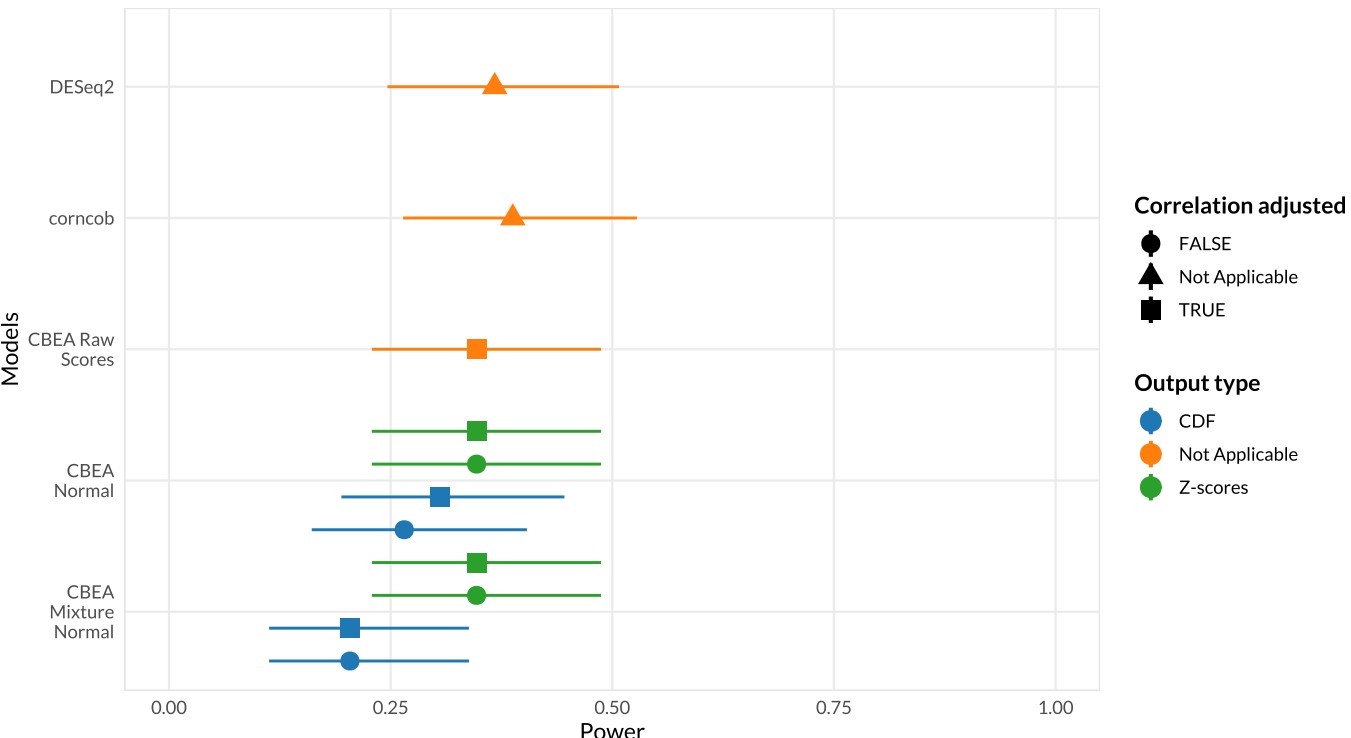

**Fig 5. Statistical power to assess phenotype relevance of inference tasks at the population level.** Power (*x*-axis) was estimated as the overall fraction of sets representing genera that are aerobic or anaerobic microbes found to be differentially enriched across sample type (supragingival or subgingival). 95% confidence intervals were computed using the Agresti-Couli approach for binomial proportions.

## Disease prediction

Since CBEA can generate informative scores that can discriminate between samples with inflated counts for a set (Fig 4), we wanted to assess whether these scores can also act as useful inputs to predictive models. In this section we assessed the predictive performance of a standard baseline random forest model [58] with different single-sample enrichment scoring methods as inputs (CBEA, ssGSEA, and GSVA). Additionally, we also compared the predictive performance of using these scores against a standard approach of using the centered log-ratio transformation (CLR) on taxon sets aggregated via abundance summations.

We fit our model to two data sets with a similar disease classification task of discriminating patients who were diagnosed with IBD (includes both Crohn's disease and ulcerative colitis) using only microbiome taxonomic composition. The two data sets represent different microbiome sequencing approaches: the Gevers et al. [57] data set uses 16S rRNA gene sequencing, while the Nielsen et al [56] data set uses whole genome shotgun sequencing.

Fig 6 illustrates the performance of our model with AUROC as the evaluation criteria. In the 16S rRNA data set, the best performing CBEA variant (CDF values computed from an unadjusted mixture normal distribution) outperforms both GSVA and ssGSEA but not the standard CLR approach. Interestingly, in the whole genome sequencing data set, CBEA outperformed CLR, but was similar in performance to GSVA. However, due to large confidence intervals, no method significantly out-performed other evaluated approaches. As such, these results indicate that, for a given pre-determined collection of sets, CBEA generated scores can be informative and provide competitive performance when acting as inputs to disease predictive models. Simulation studies (S5 Fig) showed similar results, however CBEA more

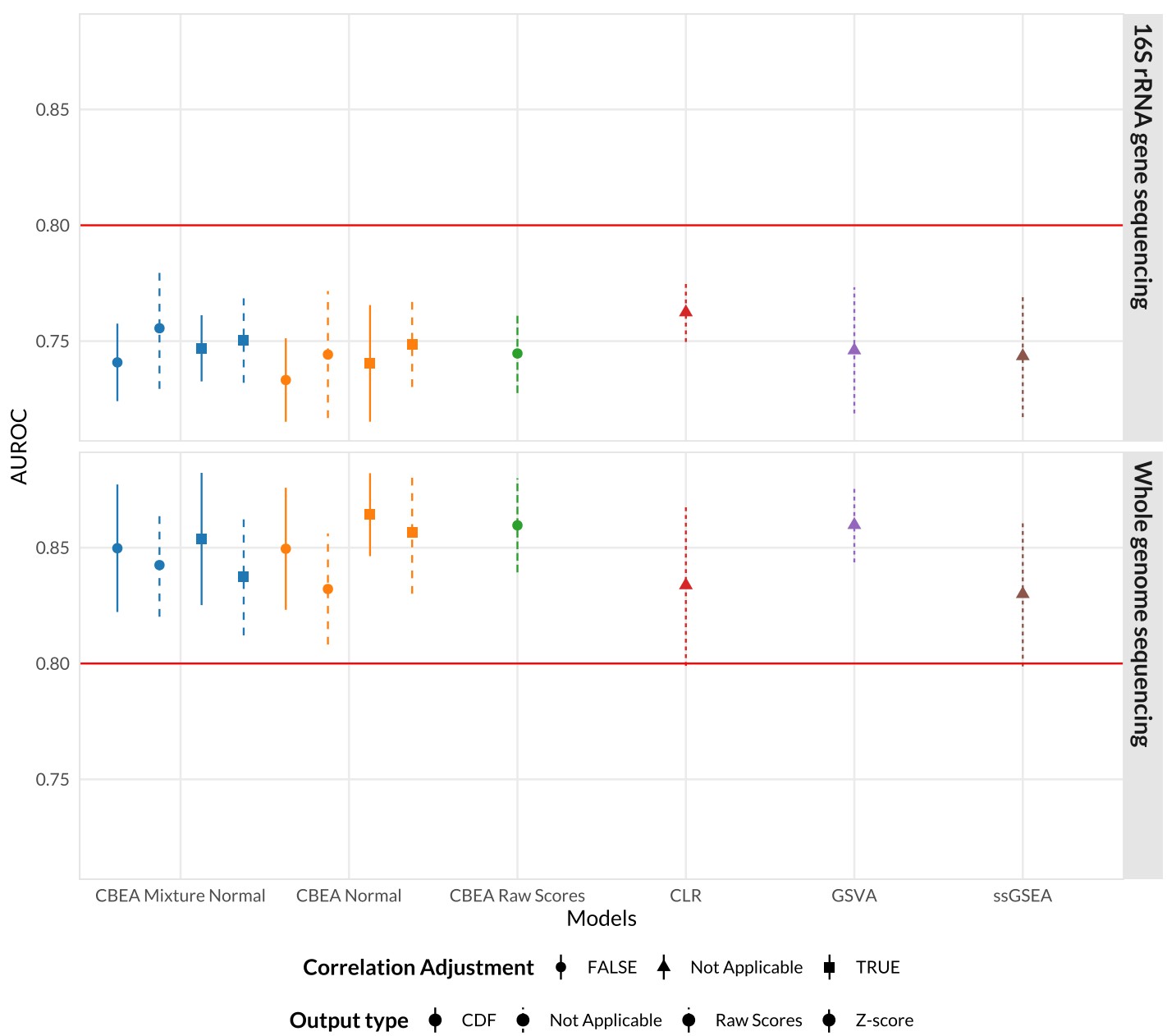

**Fig 6. Predictive performance of a naive random forest model trained on CBEA, ssGSEA, and GSVA generated scores, as well as the standard CLR approach on predicting patients with inflammatory bowel disease versus controls using genus level taxonomic profiles.** The data sets used span both 16S rRNA gene sequencing (Gevers et al. [57]) and whole-genome shotgun sequencing (Nielsen et al. [56]). CBEA performs better than GSVA and ssGSEA but not as well as CLR, with the exception of the whole genome sequencing data set.

consistently underperformed compared to CLR across all scenarios. Interestingly, the performance gap decreased with increasing sparsity levels and correlation.

## Discussion

### Inference with CBEA

CBEA is a microbiome-specific approach to generating sample-specific enrichment scores for taxonomic sets defined *a priori*. The formulation of CBEA as a comparison between taxa

within the set and its complement corresponds to the competitive null hypothesis in the gene set testing literature [27]. Since this null hypothesis is self-contained per sample, it allows users perform enrichment testing at the sample level. Additionally, in combination with a difference in means test, CBEA can also test for enrichment at the population level across case/control status similar to GSVA [15].

For single-sample analyses, we demonstrated that the CBEA approach (unadjusted with mixture normal parametric assumption) was able to control for type I error at the nominal level of 0.05 under the global null (Fig 2), while also demonstrating adequate power (Fig 4). This performance is consistent across different set sizes as well as our prior distributional fit analyses (Fig 1), in which the mixture normal displayed superior fit to the null distribution. Unfortunately, other variants of CBEA demonstrated neither good type I error control nor power. Interestingly, while the adjusted methods showed poor performance in real data evaluations (Fig 2), in simulation studies (S1 and S3 Figs) these approaches were able to control for type I error well with the trade-off of much lower power. For the population-level inference task, CBEA also performed very well. Under the permutation global null, representing genera abundance using CBEA scores in combination with Welch's t-test controls for type I error at the correct $\alpha$ threshold while also keeping respectable power. Since the population level enrichment test is equivalent to a differential abundance test using set-based features, we compared the CBEA approach against using element-wise summations with corncob [61] and DESeq2 [21] to test for set-level differential abundance. We chose DESeq2 because it is an older approach from the bulk RNA-seq literature that has strong support for usage in microbiome taxonomic data [47]. Alternately, corncob is a newer method developed specifically for microbiome taxonomic data sets, which models taxonomic counts directly using a beta-binomial distribution instead of relying on normalization via size factor estimation. We observed that using this approach resulted in an inflated type I error compared to all variants of CBEA (Fig 2), yet did not improve power (Fig 4). Results for CBEA approaches were replicated in simulation analyses, however, for corncob and DESeq2 we observed an opposite effect: in simulation experiments, both methods show good type I error control but low power (S2 and S4 Figs).

We hypothesized that the discrepancy between simulation and real data evaluations could be due to differences in our assumptions regarding the data generating process that informed our simulation schema. For the non-zero component of each taxon, we sampled from the same negative binomial distribution where designated enriched taxa were generated with inflated means (but the same dispersion). These marginals were simulated to account for block exchangable correlation within the enriched set only. This might have affected our results in two ways. First, our simulation scenario ensures that all designated non-enriched taxa are identical to each other. This is not the case for real data, because our null scenario involves randomly sampled sets that might by chance all have taxa with inflated means compared to remainder taxa. This is represented in S7 Fig, where the distribution of type I error across 500 replications is right skewed for underperforming CBEA variants, indicating that these approaches are much more sensitive compared to the Wilcoxon rank sum test or unadjusted CBEA with mixture normal distribution. Second, as described in the Introduction section, we did not consider taxon-specific biases that distort the observed relative abundance of taxa compared to true values [25]. In the context of sum-based aggregations, the resulting bias of the aggregated taxon-set is dependent on the relative abundances of the contributing taxa (Appendix I in [25]). Conceptually, this means that measurement error for a taxon-set is different across samples as the relative abundances of contributing taxa change, leading to issues when attempting to perform inference. As such, we expect methods like corncob or DESeq2, when performed on such sum aggregates in the presence of taxon-specific biases, to have inflated

type I error compared to our multiplication based approach. This also explains why conversely in simulation studies, where taxon-specific biases are absent, corncob and DESeq2 performed better.

## Downstream analysis using predictive models

The sample-level enrichment scores generated by the CBEA method can be used in downstream analyses such as disease prediction. We evaluated whether CBEA can be used to generate set-based features for disease prediction models.

We fit a basic random forest model [58] to predict continuous and binary outcomes using CBEA generated scores as inputs. Similar to our inference analysis, we compared CBEA against both ssGSEA and GSVA. Additionally, we evaluated CBEA against a standard approach where counts of a set were aggregated using sums and applied the centered log-ratio transformation (CLR). This is because CLR is considered standard practice in using microbiome variables as predictors for a model [9]. Results showed that CBEA generates scores perform well across both real data and simulation scenarios. Since predictive models consider the effect of variables jointly (and in the case of random forest, consider interactions as well), good performance indicates that CBEA scores can capture joint distribution of sets, enabling both uniset and multi-set type analyses. Comparatively, CBEA generated scores outperformed other enrichment score methods (GSVA and ssGSEA), suggesting that CBEA is more tailored for microbiome taxonomic data sets. This is consistent with our sample ranking analysis (Fig 4), showing that CBEA scores are on average more informative when used to rank samples based on their propensity to have inflated counts. However, CBEA did not outperform the CLR approach across our simulation studies, and only marginally performed better in the real data analysis with WGS data. Fortunately, in simulation studies, this performance gap between CLR and CBEA decreases with higher sparsity and correlation, especially in low effect saturation scenarios.

## Limitations and future directions

These above results demonstrate the applicability of CBEA under different data analysis scenarios. If researchers are interested in performing inference, they can decide between an unsupervised sample level approach (i.e. screen samples for enrichment of certain characteristics) or a supervised population level approach (i.e. identifying characteristics that are differentially abundant across case/control status). For the unsupervised approach, utilizing the unadjusted CBEA with the mixture normal distribution provides a good initial starting point. In the case where researchers only want to screen samples with mean-inflated taxon sets (instead of additionally detecting taxon sets with increased correlation), they can apply the adjusted approach, which can be effective at conserving type I error even for high correlation scenarios. However, the trade-off for this adjustment is power, which decreases with increasing correlation. For the supervised analysis, all CBEA variants control for type I error and provide adequate statistical power. However, using raw CBEA scores with a difference-in-means test such as Welch's t-test is preferable since is the least computationally expensive (no estimation process) while still outperforming the use of a sum-based approach with a standard differential abundance test.

Beyond inference, CBEA scores are flexible and can be useful for downstream analysis. We demonstrated that for a given number of set-based features, CBEA can produce informative scores that contribute to competitive performance of prediction models even in low signal-to-noise ratios with high inter-taxa correlation and sparsity. This is especially true for whole genome sequencing data sets, where CBEA outperfroms the standard approach of applying a CLR transformation. Researchers might find CBEA useful under situations of high sparsity

and inter-taxa correlation, or if the property of a singular covariance matrix (a byproduct of the CLR transformation [9]) is undesired. Even though we only evaluated prediction models, researchers can benchmark their own usage of CBEA for other downstream tasks such as sample ordination.

However, there are various limitations to our evaluation of CBEA. First, our simulation analysis may not capture the appropriate data-generating distributions underlying microbiome taxonomic data. There is strong evidence to suggest that our zero-inflated negative binomial distribution is representative [63], however other distributions such as the Dirichlet multinomial distribution [64] have been used in the evaluation of prior studies. More recent studies have suggested utilizing the hierarchical multinomial logistic normal distribution to model microbiome data sets [65, 66]. As such, there is space to evaluate and adapt CBEA to these different distributional assumptions that underlie the data generating process. Second, we were not able to evaluate the phenotype relevance of enrichment results as in Geistlinger et al. [41] due to limited consistent annotations for microbiome signatures in health and disease, especially those that are experimentally verified (and not just from differential abundance studies). We attempted to perform this evaluation by leveraging the gingival data set similar to [63]. However, we acknowledge that this is not a perfect solution, since the oxygen usage label of each microbe in the data set is only available at the genus level, and the difference in counts for obligate aerobes and anaerobes across the supragingival and subgingival sites might not be as clear-cut. As such, results from power analyses using this data set are only relative between the comparison methods and cannot be treated as absolute measures of power or phenotype relevance. Third, fitting the mixture normal distribution to raw CBEA scores using the expectation-maximization algorithm is difficult, as the convergence rate is slow when there is high overlap between the mixtures, resulting in a small mixing coefficient for one of the components and increased runtime (S6 Fig) [67]. In our implementation, we attempted to account for this by increasing the maximum number of iterations and relaxing the tolerance threshold. Finally, we assumed that taxa within a set are all equally associated with the outcome. This limits our ability to evaluate the performance of CBEA when only a small number of taxa within the set are associated with the outcome, or if there is variability in effect sizes or association direction of taxa within a set.

Our evaluation also showed various drawbacks of the CBEA method itself. First, inference with CBEA at the sample level is limited, and can be affected by inter-taxa correlation if users wish to detect mean-inflated sets only. Second, for downstream analyses, CBEA might not always perform better than competing methods, especially when being used to generate inputs to predictive models. We hypothesized that this might be due to the lack of fit for the underlying null distribution in high correlation settings, especially the identifiability problem associated with the estimation procedure while adjusting the mixture normal distribution. As such, we hope to refine the null distribution estimating procedure by either choosing a better distributional form, or to further constrain the optimization procedure of the mixture normal distribution by fixing the third and fourth moments.

In addition, CBEA itself did not consider other aspects of microbiome data. First, across all analyses, we relied on adding a pseudocount to ensure log operations are valid. Users can directly address this by incorporating model-based zero correction methods prior to modelling such as in [68] or [69]. However, in our simulation studies, sparsity seems to not have a significant impact on the overall performance of our approach. Second, CBEA also treated all taxa within the set as equally contributing to the set. Incorporation of taxa-specific weights (similar to PhILR [35]) could reduce the influence of outliers, such as rare or highly invariant taxa. Finally, even though for a given set of *a priori* annotations CBEA can generate useful summary scores, such values are limited in their utility if the annotations themselves are not

meaningful. As such, curating and validating sets (similar to MSigDB [12]) based on physiological or genomic characteristics of microbes [70] or their association with human disease (in beta BugSigDB https://bugsigdb.org/Main_Page) can allow for incorporating functional insights into microbiome-outcome analyses.

## Conclusion

Gene set testing, or pathway analysis, is an important tool in the analysis of high-dimensional genomics data sets; however, limited work has been done developing set based methods specifically for microbiome relative abundance data. We introduced a new microbiome-specific method to generate set-based enrichment scores at the sample level. We demonstrated that our method can control for type I error for significance testing at the sample level, while generated scores are also valid inputs in downstream analyses, including disease prediction and differential abundance.

## Supporting information

**S1 Fig. Simulation results for type I error evaluation for CBEA sample-level inference.** Type I error rate ($y$ axis) was estimated for each approach across data sparsity levels ($x$ axis) across different set sizes (horizontal) and inter-taxa correlation within the set (vertical). We compared variatns of CBEA against a Wilcoxon rank sum test at $\alpha$ of 0.05. For each scenario, a data set of 10,000 samples (equivalent to 10,000 hypotheses) was utilized. Confidence bounds were obtained using Agresti-Couli approach.
(EPS)

**S2 Fig. Simulation results for type I error evaluation for CBEA population-level inference.** Type I error ($y$-axis) was estimated as the average proportion of sets with significant enrichment at 0.05 across 10 replications per simulation condition under the global null. Error bars were estimated using standard errors computed across 10 replicated data sets. Performance was evaluated across different sparsity ($x$-axis) and inter-taxa correlation levels. For CBEA methods, enrichment analysis was performed using a Welch's t-test across case/control status with single-sample scores representing set-based features generated by CBEA (across different output types and distributional assumptions). For corncob and DESeq2, set-based features were constructed using element-wise summations.
(EPS)

**S3 Fig. Simulation results for phenotype relevance evaluation for CBEA sample-level inference. (A)** demonstrate statistical power ($y$-axis) across different data sparsity levels ($x$-axis) and power **(B)** for differential abundance test across different parametric simulation scenarios. For CBEA methods, differential abundance analysis was performed using a difference in means test (either Wilcoxon rank-sum test or Welch's t-test) across case/control status using single-sample scores generated by CBEA (across different output types and distributional assumptions). CBEA associated methods demonstrated similar type I error to conventional differential abundance analysis methods but with more power to detect differences even at small effect sizes.
(EPS)

**S4 Fig. Simulation results for phenotype relevance evaluation for CBEA population-level inference.** Power ($y$-axis) was estimated as the average proportion of sets correctly identified as significantly enriched (at 0.05) across 10 replications per simulation condition under the global null. Error bars were estimated using standard errors computed across 10 replicated data sets. Performance was evaluated across different sparsity ($x$-axis) and inter-taxa

correlation levels. For CBEA methods, enrichment analysis was performed using a Welch's t-test across case/control status with single-sample scores representing set-based features generated by CBEA (across different output types and distributional assumptions). For corncob and DESeq2, set-based features were constructed using element-wise summations.
(EPS)

**S5 Fig. Simulation results for predictive pefromance evaluation for CBEA.** Predictive performance of a random forest model (with no hyperparameter tuning) trained on set-based features as inputs. Methods to generate these features include CBEA, ssGSEA, GSVA, and the CLR transformation applied on sum-aggregated sets. Simulation data was generated across different levels of data sparsity, inter-taxa correlation, effect saturation, and signal-to-noise ratio. Panel **(A)** presents performance on a regression task using RMSE (root mean squared error) as the evaluation measure. Panel **(B)** presents performance on a classification task with AUROC as the evaluation measure.
(EPS)

**S6 Fig. Runtime performance.** Overall runtime of CBEA under different parameters for a data set of 500 samples, 800 taxa (40 sets of size 20 each). This data set was generated via simulations.
(EPS)

**S7 Fig. Distribution of type I error values across all replications in real data random set evaluations for CBEA inference at the sample-level.** Density ($y$-axis) for type I error values ($x$-axis) of each evaluated approach for sample-level inference using real data across 500 replications. Here, type I error was estimated as the proportion of samples where a randomly sampled set of different sizes where identified to be statistically significant at $p$-value threshold of 0.05.
(EPS)

**S1 File. Supplemental derivations.** Includes additional details on addressing variance inflation due to correlation in CBEA, simulation analyses, and run-time performance.
(PDF)

## Acknowledgments

The authors thank Becky Lebeaux, Modupe Coker, Erika Dade, Jie Zhou, and Weston Viles for insightful comments and suggestions that greatly improved the paper.

## Author Contributions

**Conceptualization:** Quang P. Nguyen, Anne G. Hoen, H. Robert Frost.

**Formal analysis:** Quang P. Nguyen.

**Funding acquisition:** Anne G. Hoen, H. Robert Frost.

**Investigation:** Quang P. Nguyen.

**Methodology:** Quang P. Nguyen, H. Robert Frost.

**Resources:** Quang P. Nguyen.

**Software:** Quang P. Nguyen.

**Supervision:** Anne G. Hoen, H. Robert Frost.

**Validation:** Quang P. Nguyen.

**Visualization:** Quang P. Nguyen.

**Writing – original draft:** Quang P. Nguyen.

**Writing – review & editing:** Quang P. Nguyen, Anne G. Hoen, H. Robert Frost.

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
