## [Decision Letter · Decision Letter 0]

22 Oct 2021

Dear Dr. Frost,

Thank you very much for submitting your manuscript "cILR: Competitive isometric log-ratio for taxonomic enrichment analysis" for consideration at PLOS Computational Biology.

As with all papers reviewed by the journal, your manuscript was reviewed by members of the editorial board and by several independent reviewers. In light of the reviews (below this email), we would like to invite the resubmission of a significantly-revised version that takes into account the reviewers' comments.

We cannot make any decision about publication until we have seen the revised manuscript and your response to the reviewers' comments. Your revised manuscript is also likely to be sent to reviewers for further evaluation.

Sincerely,

Nicola Segata

Associate Editor

PLOS Computational Biology

Kiran Patil

Deputy Editor

PLOS Computational Biology

Reviewer's Responses to Questions

**Comments to the Authors:**

Reviewer #1: 

1 Summary

=========

  In this manuscript Nguyen et al propose cILR for a set-enrichment-like

  analysis of microbiome sequencing data. This is a scale invariant

  alternative to the more standard (and problematic) approach of

  applying GSEA to DESeq2 output or other such approaches. Overall its a

  nice idea and is fairly well executed. The statistical analysis is

  largely appropriate and it could be a nice contribution. My largest

  comments relate to the lack of precision in the authors' writing, the

  extent of the authors claims, and the authors handling of zeros and

  count variation.

2 Detailed Comments

2.1 Precision of Writing:

~~~~~~~~~~~~~~~~~~~~~~~~~

  + The authors repeatedly state this data is "compositional" or

    "strictly compositional". First, what is "strictly compositional" is

    there non-scrictly compositional data? Second, the data is clearly

    not compositional; this is a cliche that has been amplified in the

    literature and is incorrect. The data is count data, it has zeros

    and is integer valued, both of those features are in strict contrast

    to the standard definition of compositional data (i.e., positive

    multivariate continuous data that sums to a constant value and

    typically is an open set excluding zero to avoid issues with

    log-ratio transforms). This distinction is non-trivial as the direct

    application of log-ratio transforms to this data is poorly motivated

    in this case and fraught with problems (See later comments on

    handling of the data).

  + What the authors propose is not an ILR transform. Unless I am

    mistaken, there is no constraint on the matrix A such that the the

    coordinate system is cartesian with an orthonormal basis. In fact,

    if k does not equal p-1 then it cannot possibly be isomorphic let

    alone isometric with respect to the Aitchison metric. Unless I am

    mistaken, the authors should change the name of their method and

    modify their discussion to be more accurate. I would relate this

    method is not an ILR transform but it is very similar to phylofactor

    which takes a similar approach (in phylofactor set membership is

    dictated by the topology of the tree).

  + The authors state that the "data is zero-inflated" this is another

    cliche that I would encourage the authors to remove. Zero-inflation

    is a particular family of models for these zeros not a objective

    characteristic of the data. Simply saying there are many zeros would

    likely suffice in this article. They could argue that the data

    generating mechanism is well represented by a zero-inflation process

    but this has been called into question (see Silverman et al. Naught

    all zeros in sequence count data are the same.)

  + The authors state that the data is compositional because the number

    of reads obtained is constrained by the sequencing instrument. Would

    an instrument that didn't have this constraint lead to

    "non-compositional data"? This seems unlikely. For example standard

    equimolar pooling protocols explicitly dilute concentrated DNA from

    each sample to try to equalize sequencing depth. Its not just an

    issue of the sequencer. Even sampling from an environment (e.g.,

    taking 5 grams of stool from a larger stool sample) looses the

    notion of absolute abundance.

  + The GSEA method cited on line 51 is not a random-walk like

    statistic. I think it may be a brownian bridge but its constrained

    to be zero at either end -- not a random walk.

  + Between lines 73 and 85 the authors do not properly motivate the

    multiplicative rather than additive amalgamation. They mention the

    downsides of the "naive summation-based method" but this is unclear.

    From later in the manuscript I gather that this statement reflects

    the perturbation invariance of multiplicative amalgamation: given

    that some have argued that measurement bias can be modeled as a

    constant compositional perturbation. This needs to be made explicit.

    There is no inherent downside of summation (i.e., additive

    amalgamation) - its a modeling choice and it is not "naive".

  + The authors mention "adjusting for correlation" multiple times

    throughout the manuscript yet the motivation is not properly

    clarified. The best I can guess is that they are saying that they

    need to modify the null-hypothesis to account for a trivial case

    where something looks differential expressed or set enriched when

    really its just due to the correlation structure between taxa. That

    said, I think there are many potential sources of confusion that the

    authors should clarify. Couldn't set enrichment be reflected in

    those correlations? Isn't the correlations actually a non-trivial

    part of what the authors are trying to model? In other words if a

    set of microbes is highly correlated wouldn't that be a sign that

    that set is potentially enriched or de-enriched? I don't think I

    understand this point completely but I think it is likely

    non-trivial. I would encourage the authors to clarify the role of

    correlation.

  + On lines 167-170 the authors state that since the cILR are not

    orthogonal a correlation can exist between cILR aggregated

    variables. This is misleading there can be correlation whether or

    not the cILR's are orthogonal or not, orthogonality and a lack of

    correlation are separate concepts.

  + Citation on line 187. I don't see how this paper supports this

    statement. Egozcue et al. take a almost purely mathematical approach

    as far as I can tell do not discuss central limit theorems or other

    things that are implied by the authors statement. If I remember

    correctly the relevant citations are authored by Aitchison while I

    cannot remember them exactly.

  + On line 536 the authors mention "inflated counts", I have no idea

    what this means.

  + Lines 572 to 582. I don't understand how this hypothesis makes

    sense. How does taxa-specific bias relate to the performance of

    DESeq2 or corncob? The writing here is poor. Also, I am not sure how

    this could be, are you not basing the gold-standard truth off of

    permuted data which you know has no signal? This permutation would

    maintain the measurement bias... as a result it would seem the data

    does not support this hypothesis. I expect I am missing something.

2.2 Modeling Choices

~~~~~~~~~~~~~~~~~~~~

  + As far as I can tell the authors do not state how they are handling

    zeros. This is a non-trivial methodologic detail especially if they

    are simply taking log-ratio transforms of count data. To what extend

    is the non-normality of the authors results simply a product of

    directly transforming count data without accounting for the variance

    of the counts. For example, count data typically have a mean

    variance relationship that seems largely ignored by the authors

    approach. Moreover, there has been a number of advances in

    compositional modeling of microbiome focusing on Multinomial

    logistic normal models that are not addressed by the authors. In

    fact in light of the availability of these methods the authors

    modeling of these counts seems sub-par.

2.3 Unsubstantiated Claims

~~~~~~~~~~~~~~~~~~~~~~~~~~

  There are a number of unsubstantiated claims where the language needs

  to be altered to be more precise.

  + Line 493: "These results demonstrate that cILR generated scores are

    informative features in disease prediction tasks." No. These results

    demonstrate that cILR COULD be informative features in disease

    predictions tasks. I am not convinced that these are even useful for

    the case-studies shown in this manuscript let alone other tasks.

    Moreover, the comparison methods ssGSEA and GSVA seem like odd

    choices. Are the authors only using methods that can take set-based

    features? This does not account for the potential that the chosen

    sets are not informative. The later seems like an important case to

    establish the motivation of the current work.

  + Line 535-537. The authors show that their model displays Type 1

    error control on a set of of simulated datasets. They make some

    claims about false-discovery control on real data on lines 397-406

    but I really don't follow how they know what is non-random or random

    on this dataset. It seems like they have a strong hypothesis about

    aerobic vs. anaerobic but that hypothesis seems too weak to serve as

    a gold-standard reference. Overall these claims are unsubstantiated.

    I would emphasize that any claim saying a model can be trusted is

    suspect and bordering on overtly false -- any model can fail and

    nearly all models are misspecified there are times at which a model

    may be useful but that iss about it. No model can be globally

    trusted.

2.4 Other Comments on Clarity

~~~~~~~~~~~~~~~~~~~~~~~~~~~~~

  + The writing after line 215 lacks detail. I kept waiting for a

    methods section to answer some of my questions (e.g., how was mu or

    phi chosen in equation 3) but these don't seem to be listed

    anywhere. Details in the remaining parts of the manuscript are

    inconsistently given or vague. e.g., Line 249 "all sample sizes were

    set to 10,000". Do you mean sequencing depth? Number of reads?

    Number of technical replicates? What is this referring to?

  + The writing after line 215 is hard to read. In part this relates to

    the lack of detail but I think it also stems from the fact that the

    manuscript starts being written in triplicate for Single Sample

    Enrichment, Differential Abundance Analysis, and then Prediction. It

    makes the paper repetitive and hard to follow. Further, figures are

    repetitive and poorly labeled so its hard, at a glance, to figure

    out what figure links to what part of the paper. I have never seen a

    discussion written in the parts but this just adds to the feeling

    that this is just a paper written in triplicate without a coherent

    message beyond the initial idea which ends around line 215. Further

    it was not clear from reading the introduction that the paper would

    be organized like this; some warning in the introduction may help a

    bit. In fact, it was not even clear what the distinction between

    single-sample enrichment and differential abundance was from the

    introduction. In addition, this notation is non-standard. Typically

    enrichment (e.g., as used in gsEa) refers to essentially

    differential expression but for sets of genes (i.e., it is typically

    a comparison between groups). This makes the "single-sample

    enrichment" terminology confusing.

Reviewer #2: * Summary:

Nguyen et al. present a new method for taxonomic enrichment analysis of microbiome data based on an isometric log-ratio transformation of compositional and the competitive null hypothesis borrowed from the gene set enrichment literature. The main strengths are a well-written and structured manuscript, a solid statistical and analytical foundation of the method, and a thorough evaluation of the method on simulated and real datasets. The main weaknesses are installation issues with the R companion package, a lack of adaptation of existing standards for the benchmarking of gene set enrichment methods, and a number of theoretical considerations with the use of a competitive null hypothesis for enrichment testing.

* Major:

** Installation:

Using a recent R installation (R.4.1.0) and Bioconductor installation (3.13), I was not able to install the package. The error message and my session info is included below. The method looks useful for the community and I would strongly encourage a Bioconductor submission (or at least a CRAN submission) of the package to ensure that the package passes R CMD build, check, and install in a continuous integration setup.

** Adapting standards for the benchmarking of enrichment methods:

Geistlinger et al. (doi: 10.1093/bib/bbz158) has recently introduced an extensible framework for reproducible benchmarking of enrichment methods based on defined criteria for applicability, gene set prioritization and detection of relevant processes. This setup consists of compendia of curated and standardized datasets and a number of criteria that apply as-is also for new enrichment methods in the microbiome data realm (such as runtime, proportion of rejected null hypotheses, behavior on permuted sample labels and random gene sets). Although I would really like to commend the authors for using curated and standardized datasets from curatdMetagenomicData and HMP16SData, the authors then proceed with the practice of self-assessment over various scenarios which is typically difficult to transport and apply for new methods. Being one of the first methods for enrichment analysis in the microbiome realm (but very likely not the last one), the paper has the opportunity to very early on set the baseline for how new enrichment methods in the microbiome space should be evaluated building on lessons learned in the gene set enrichment literature. This could be achieved (a) clearly communicating the existence of such standards, (b) adapting existing standards where possible, and (c) to point out where adaption of such standards would require further work, as there might well be criteria that do not straightforward translate from gene set enrichment to taxon set enrichment.

** On the use of the competitive null hypothesis for enrichment testing:

The authors demonstrate that cILR controls for type I error even under high sparsity and high inter-taxa correlation. However, it has been pointed out that strict type I error rate control might not be a desirable feature for enrichment methods (Goeman and Buhlman, 2009; Wu and Smyth, 2012; Geistlinger et al. 2021). Gene set enrichment analysis is an exploratory process, not a confirmatory, diagnostic process, where strict type I error control augments the lack in power which is well documented for competitive enrichment testing (Goeman and Buhlman, 2009; Wu and Smyth, 2012; Geistlinger et al. 2021) and as the authors demonstrate in their own evaluations. Furthermore, Geistlinger et al. 2021 (Figure 4 therein) has demonstrated that despite controlling the type I error rate, methods might demonstrate widely different rejection rates on real datasets. It is in this context noteworthy that the authors of Camera (Wu and Smyth, 2012), which deliberately abandons strict type I error control by default to compensate for the apparent lack in power of competitive methods.

> devtools::install_github("qpmnguyen/teaR")

checking for file ‘/private/var/folders/q9/zp53ds5x6s9g62l77whflgyc0000gn/T/RtmpaSffST/remotesc96e405f9f91/qpmnguyen-teaR-ddb✔ checking for file ‘/private/var/folders/q9/zp53ds5x6s9g62l77whflgyc0000gn/T/RtmpaSffST/remotesc96e405f9f91/qpmnguyen-teaR-ddbc50c/DESCRIPTION’

─ preparing ‘teaR’:

✔ checking DESCRIPTION meta-information ...

─ cleaning src

─ checking for LF line-endings in source and make files and shell scripts

─ checking for empty or unneeded directories

Omitted ‘LazyData’ from DESCRIPTION

─ building ‘teaR_0.99.0.tar.gz’

Installing package into ‘/Library/Frameworks/R.framework/Versions/4.1/Resources/release’

(as ‘lib’ is unspecified)

* installing *source* package ‘teaR’ ...

** using staged installation

** libs

clang++ -I"/Library/Frameworks/R.framework/Resources/include" -DNDEBUG -I'/Library/Frameworks/R.framework/Versions/4.1/Resources/release/Rcpp/include' -I/usr/local/include -fPIC -Wall -g -O2 -c RcppExports.cpp -o RcppExports.o

clang++ -I"/Library/Frameworks/R.framework/Resources/include" -DNDEBUG -I'/Library/Frameworks/R.framework/Versions/4.1/Resources/release/Rcpp/include' -I/usr/local/include -fPIC -Wall -g -O2 -c cilr.cpp -o cilr.o

cilr.cpp:26:41: error: expected expression

if (std::any_of(X.begin(), X.end(), [](double i){return i <= 0;})){

^

1 error generated.

make: *** [cilr.o] Error 1

ERROR: compilation failed for package ‘teaR’

* removing ‘/Library/Frameworks/R.framework/Versions/4.1/Resources/release/teaR’

Warning message:

In i.p(...) :

installation of package ‘/var/folders/q9/zp53ds5x6s9g62l77whflgyc0000gn/T//RtmpaSffST/filec96e620d5765/teaR_0.99.0.tar.gz’ had non-zero exit status

> sessionInfo()

R version 4.1.0 (2021-05-18)

Platform: x86_64-apple-darwin17.0 (64-bit)

Running under: macOS Big Sur 10.16

Matrix products: default

BLAS: /Library/Frameworks/R.framework/Versions/4.1/Resources/lib/libRblas.dylib

LAPACK: /Library/Frameworks/R.framework/Versions/4.1/Resources/lib/libRlapack.dylib

locale:

[1] en_US.UTF-8/en_US.UTF-8/en_US.UTF-8/C/en_US.UTF-8/en_US.UTF-8

attached base packages:

[1] stats graphics grDevices utils datasets methods base

loaded via a namespace (and not attached):

[1] rstudioapi_0.13 magrittr_2.0.1 usethis_2.0.1

[4] devtools_2.4.2 pkgload_1.2.1 R6_2.5.0

[7] rlang_0.4.11 fastmap_1.1.0 tools_4.1.0

[10] pkgbuild_1.2.0 sessioninfo_1.1.1 cli_3.0.1

[13] withr_2.4.2 ellipsis_0.3.2 remotes_2.4.0

[16] rprojroot_2.0.2 lifecycle_1.0.0 crayon_1.4.1

[19] processx_3.5.2 BiocManager_1.30.16 purrr_0.3.4

[22] callr_3.7.0 fs_1.5.0 ps_1.6.0

[25] testthat_3.0.4 curl_4.3.2 memoise_2.0.0

[28] glue_1.4.2 cachem_1.0.5 compiler_4.1.0

[31] desc_1.3.0 prettyunits_1.1.1

**Have the authors made all data and (if applicable) computational code underlying the findings in their manuscript fully available?**

Reviewer #1: Yes

Reviewer #2: Yes

PLOS authors have the option to publish the peer review history of their article (what does this mean?). If published, this will include your full peer review and any attached files.

Reviewer #1: No

Reviewer #2: No
---

## [Decision Letter · Decision Letter 1]

8 Mar 2022

Dear Dr. Frost,

Thank you very much for submitting your manuscript "CBEA: Competitive balances for taxonomic enrichment analysis" for consideration at PLOS Computational Biology. As with all papers reviewed by the journal, your manuscript was reviewed by members of the editorial board and by several independent reviewers. The reviewers appreciated the attention to an important topic. Based on the reviews, we are likely to accept this manuscript for publication, providing that you modify the manuscript according to the review recommendations.

Sincerely,

Nicola Segata

Associate Editor

PLOS Computational Biology

Kiran Patil

Deputy Editor

PLOS Computational Biology

[LINK]

Reviewer's Responses to Questions

**Comments to the Authors:**

Reviewer #1: This response was well thought out and addresses my major concerns.

Reviewer #2: My comments regarding the manuscript text have been satisfactorily addressed. The authors also indicate that they started a Bioconductor submission under https://github.com/Bioconductor/Contributions/issues/2449. However, the submission under the provided link has been closed due to inactivity / revisions performed to the manuscript and implementation. An initial review by a reviewer from the Bioc core team indicates that the package is in good shape and would only need minor modifications for acceptance in Bioconductor. As I assume this new method to be a useful contribution for the community, I would thus strongly encourage to follow through with the Bioconductor submission and include a link to the Bioconductor package in the manuscript (once the package has passed review and is accepted in Bioconductor).

**Have the authors made all data and (if applicable) computational code underlying the findings in their manuscript fully available?**

Reviewer #1: None

Reviewer #2: Yes

PLOS authors have the option to publish the peer review history of their article (what does this mean?). If published, this will include your full peer review and any attached files.

Reviewer #1: No

Reviewer #2: No

Figure Files:

Data Requirements:

Reproducibility:

References:

---

## [Decision Letter · Decision Letter 2]

8 Apr 2022

Dear Dr. Frost,

We are pleased to inform you that your manuscript 'CBEA: Competitive balances for taxonomic enrichment analysis' has been provisionally accepted for publication in PLOS Computational Biology.

Best regards,

Nicola Segata

Associate Editor

PLOS Computational Biology

Kiran Patil

Deputy Editor

PLOS Computational Biology

Reviewer's Responses to Questions

**Comments to the Authors:**

Reviewer #2: I have no further comments.

**Have the authors made all data and (if applicable) computational code underlying the findings in their manuscript fully available?**

Reviewer #2: Yes

PLOS authors have the option to publish the peer review history of their article (what does this mean?). If published, this will include your full peer review and any attached files.

Reviewer #2: No

---

## [Editor Report · Acceptance letter]

2 May 2022

PCOMPBIOL-D-21-01648R2 

CBEA: Competitive balances for taxonomic enrichment analysis

Dear Dr Frost,

I am pleased to inform you that your manuscript has been formally accepted for publication in PLOS Computational Biology. Your manuscript is now with our production department and you will be notified of the publication date in due course.

With kind regards,

Olena Szabo
